# Effect and Correlation of *Rosa roxburghii* Tratt Juice Fermented by *Lactobacillus paracasei* SR10-1 on Oxidative Stress and Gut Microflora Dysbiosis in Streptozotocin (STZ)-Induced Type 2 Diabetes Mellitus Mice

**DOI:** 10.3390/foods12173233

**Published:** 2023-08-28

**Authors:** Maoyang Wei, Dandan Feng, Yulong Zhang, Yunyang Zuo, Jiuchang Li, Ling Wang, Ping Hu

**Affiliations:** School of Liquor and Food Engineering, Guizhou University, Guiyang 550025, China; maoyangwei2021@126.com (M.W.); danfeng17885906867@163.com (D.F.); zhangy1340@126.com (Y.Z.); 18748725666@163.com (Y.Z.); lijiuchang307@163.com (J.L.); wl870728641@163.com (L.W.)

**Keywords:** *Rosa roxburghii* Tratt, type 2 diabetes mellitus, short-chain fatty acid, gut microbiota

## Abstract

*Rosa roxburghii* Tratt (RRT) is a kind of excellent fruit, with many healthy functions. RRT fruit dietary interventions have demonstrated a remarkable potential to prevent type 2 diabetes mellitus (T2DM). In the present study, the effects of *Lactobacillus paracasei* SR10-1 fermented RRT juice (FRRT) on the oxidative stress, short-chain fatty acids (SCFAs), and gut microbiota in T2DM mice induced by high-sugar and high-fat diets and streptozotocin (STZ) were investigated using GC–MS and 16S rRNA gene sequencing. The results showed that medium-dose FRRT intervention resulted in significantly decreased levels of TG, TC, LDL-C, BUN, creatinine, and MDA (*p* < 0.05) and significantly increased levels of HDL-C, GSH-PX, CAT, and SOD of T2DM mice (*p* < 0.05). The levels of acetic acid, propionic acid, butyric acid, and isovaleric acid were significantly increased, by 142.28%, 428.59%, 1968.66%, and 81.04% (*p* < 0.05), respectively. The relative abundance of Firmicutes, Lachnospiraceae, Verrucomicrobiaceae, *Akkermansia*, and *Allobaculum* was significantly increased (*p* < 0.05), and the relative abundance of Proteobacteria, Enterobacteriaceae, Veillonellaceae, *Phascolarctobacterium*, and *Klebsiella* was significantly decreased (*p* < 0.05). Correlation analysis showed that *Phascolarctobacterium* was significantly negatively correlated with weight (*p* < 0.05), SOD (*p* < 0.01), CAT (*p* < 0.05), and T-AOC (*p* < 0.05). *Akkermansia* was significantly negatively correlated with weight (*p* < 0.05). Conclusively, medium-dose FRRT potentially improved T2DM by reversing dyslipidemia, decreasing oxidative stress, increasing SCFAs, and regulating gut microbiota composition. The medium-dose FRRT may serve as a novel T2DM dietary strategy to prevent T2DM.

## 1. Introduction

Type 2 diabetes mellitus (T2DM), a metabolic disease characterized by hyperglycemia and insulin resistance that accounts for more than 90% of all cases of diabetes, is a growing global health problem [1,2,3]. The onset of T2DM is associated with insufficient insulin secretion or insulin resistance. Modern pharmacological studies have shown that gut microbiota play an essential role in the onset and development of T2DM, and are increasingly recognized as a critical factor in human health [4,5,6,7]. Many probiotic-fermented fruit juices have been used to regulate gut microbiota with remarkable effects [8,9]. Gao et al. found that fermented *Momordica charantia* L. juice had better effects on hyperglycemia, hyperinsulinemia, hyperlipidemia, and oxidative stress than unfermented *Momordica charantia* L. juice in diabetic rats [10]. Probiotic-fermented fruit juices can prevent obesity and modulate the gut microbiota, and the prevention of T2DM by probiotic-fermented fruit juices will bring substantial benefits to patients. Foods with antidiabetic properties have attracted increasing attention [1,11,12].

*Rosa roxburghii* Tratt (RRT) fruit is a medicinal plant and traditional food in Southwest China, which is rich in vitamin C, superoxide dismutase (SOD), and phenolic compounds, with vitamin C content as high as 1300–3500 mg/100 g, and is known as the “King of Vitamin C” [13,14,15]. RRT fruit and its active compounds have been proven to have antioxidant, anticancer, anti-atherosclerotic, antihypertensive, antidiabetic, and anti-inflammatory properties [13,16,17,18,19,20]. Polyphenols in RRT fruit have shown the improvement of T2DM [21,22], which provides a basis for the improvement of T2DM by fermented RRT fruit juice (FRRT). FRRT can ameliorate intestinal dysbacteriosis, metabolic disorders, and dyslipidemia [23].

The probiotic fermentation of RRT juice as a convenient and fast functional food has attracted much attention. Our previous study demonstrated that the fermentation by *Lactobacillus paracasei* SR10-1 significantly increased the hypoglycemic and hypolipidemic effects of RRT fruit juice (RRTJ) [24]. Therefore, the aim of this study is to evaluate the effects of FRRT on dyslipidemia, oxidative stress, short-chain fatty acids (SCFAs), and gut microbiota in streptozotocin (STZ)-induced T2DM mice.

## 2. Materials and Methods

### 2.1. FRRT Preparation

According to the team’s previous research [25], fresh RRT fruits (planted in Guizhou, China) were pulped, filtered, and aseptically filled into RRTJ after ultra-high temperature instant sterilization. *Lactobacillus paracasei* SR10-1 (CCTCC No.: M2016527) was isolated from the sour meat of the Dong nationality in China. RRTJ (vitamin C: 1724.75 ± 14.86 mg/100 g, polyphenol: 323.44 ± 0.69 mg/L) was mixed with 4 volumes of distilled water, 9.7% sucrose was added, pasteurized, inoculated with 4.3% *Lactobacillus paracasei* SR10-1 (9 log CFU/mL), and fermented at 32 °C for 64.5 h to obtain FRRT (vitamin C: 375.65 ± 7.39 mg/100 g, polyphenol: 27.66 ± 0.09 mg/L).

### 2.2. Animal Experiment

Male Kunming mice (SPF, 4 weeks old, 25 ± 3 g) were purchased from Liaoning Changsheng Biotechnology Co., Ltd. (License No.: SCXK (Liao) 2020-0001, Shenyang, China). The experiment was approved by the Animal Experiment Ethics Committee of Guizhou University (Approval No.: EAE-GZU-2021-E006). One hundred twenty mice were housed in a well-ventilated room at a temperature of 25 ± 2 °C, with relative humidity 50 ± 5%, and exposed to 12 h/12 h day–night cycles. After the adaptive feeding of the mice was over, they were randomly divided into ten groups, each consisting of 12 mice: (1) the normal group (N, normal saline, 10 mL/kg), (2) the normal control group (FN, FRRT, 10 mL/kg), (3) the model group (M, normal saline, 10 mL/kg), (4) the positive control group (P, metformin hydrochloride, 10 mL/kg), (5) the vitamin C control group (V, vitamin C, 10 mL/kg), (6) the *Lactobacillus paracasei* SR10-1 control group (R, *Lactobacillus paracasei* SR10-1 suspension, 10 mL/kg), (7) the *Rosa roxburghii* Tratt fruit juice group (CL, RRTJ, 10 mL/kg), (8) the low-dose FRRT group (FL, FRRT, 5 mL/kg), (9) the medium-dose FRRT group (FM, FRRT, 10 mL/kg), and (10) the high-dose FRRT group (FH, FRRT, 15 mL/kg). The N and FN groups were fed with normal feed (Beijing Hfk Biosicence Co., Ltd., Beijing, China). The M, P, V, R, CL, FL, FM, and FH groups were fed with high-fat and high-sugar feed (58.5% normal feed, 20% sucrose, 10% egg yolk powder, 10% fat, 1% cholesterol, and 0.5% sodium deoxycholate), and the mice were fed and watered freely and weighed once a week. After 4 weeks of feeding, the mice were fasted for 12 h; STZ (70 mg/kg body weight) dissolved in 0.1 mol/L sodium citrate buffer was injected intraperitoneally in the M, P, V, R, CL, FL, FM, and FH groups, and normal saline (70 mg/kg body weight) was injected intraperitoneally in the N and FN groups. After the intraperitoneal injection, feeding was continued for 1 week, and mice were fasted for 12 h. Tail blood was collected with a glucometer to determine fasting blood glucose (FBG). FBG ≥ 11.1 mmol/L is the success standard for the T2DM model [26]. After successful T2DM modeling, different samples and doses of interventions were used according to the requirements of each group. The concentrations of metformin hydrochloride and vitamin C were 100 mg/kg, and the suspension concentration of *Lactobacillus paracasei* SR10-1 was 9 log CFU/mL. The dose of oral administration was adjusted according to the body weight of the mice, and the dose was given at the same time every day. The intervention period was 4 weeks. FBG and body weight were recorded once a week (Figure 1).

### 2.3. Sample Collection

On the day before the end of the experiment, the mice were placed in a sterile mouse cage, and fresh feces samples of mice were collected in sterile Eppendorf tubes. On the last day, the blood samples were collected from the eyeballs of mice after fasting for 12 h and kept in an anticoagulant tube containing heparin sodium. The blood samples were centrifuged at 3500 rpm and 4 °C for 15 min, and the upper serum was collected. The mice were dissected under sterile conditions, and the livers and kidneys were weighed. A portion of the tissue was rinsed with pre-cooled saline water at 4 °C and placed in a 4% polyoxymethylene solution for histopathological examination. The remainder of the tissue was stored at −80 °C. The cecum contents of mice were collected in sterile Eppendorf tubes. All samples were stored at −80 °C until analysis.

### 2.4. Oral Glucose Tolerance Test

In the fourth week, mice were fasted for 12 h and orally administered 2.00 g/kg BW of glucose. Then, their blood was taken by clipping the tails to measure blood glucose levels at 0 min, 30 min, 60 min, and 120 min. The area under the blood glucose curve (AUC) was calculated from Equation (1).
(1)$$AUC(hmmol/L)=\frac{0.5A+B+C+0.5D}{2}$$

In Equation (1), *A*, *B*, *C*, and *D* represent the blood glucose values at 0 min, 30 min, 60 min, and 120 min, respectively, after oral gavage of glucose solution.

### 2.5. Organ Coefficient

The organ coefficient is mainly used as an index to assess whether the feeding of FRRT causes the abnormalities of the liver, spleen, kidney, and heart of mice. An increase in the organ coefficient indicates congestion, edema, or hypertrophy of the organ, and a decrease in the organ coefficient indicates atrophy of the organ and other degenerative changes. The organ coefficient was calculated based on Equation (2).
Organ coefficient (%) = (mouse organ weight/mouse body weight) × 100%(2)

### 2.6. Biochemical Analysis of Liver and Kidney

Commercial kits (Nanjing Jiancheng Bioengineering Institute, Nanjing, China) were used to determine the levels of triacylglycerol (TG), total cholesterol (TC), high-density lipoprotein cholesterol (HDL-C), and low-density lipoprotein cholesterol (LDL-C) in the liver and the levels of blood urea nitrogen (BUN) and creatinine in the kidney.

### 2.7. Histopathological Analysis

The liver and kidney tissues were fixed with a 4% polyoxymethylene solution, dehydrated, embedded in paraffin wax, sliced, and stained with hematoxylin and eosin (H&E), respectively. Samples were then observed with an optical microscope.

### 2.8. Determination of Antioxidant Enzyme Activities

Commercial kits (Nanjing Jiancheng Bioengineering Institute, Nanjing, China) were used to determine the levels of glutathione peroxidase (GSH-PX), catalase (CAT), superoxide dismutase (SOD), total antioxidant capacity (T-AOC), and malondialdehyde (MDA) in the serum.

### 2.9. Analysis of Short-Chain Fatty Acids

Saturated sodium chloride solution, 3 mol/L HCl, and ether, with a volume of 0.4, 0.05, and 0.5 mL, respectively, were added to a 50 mg fecal sample for extraction. Mixtures were subsequently shaken for 10 min at 4 °C and centrifuged for 10 min at 12,000 rpm at 4 °C to obtain the supernatant. The CaSO_4_ was added to the supernatant to absorb the water, and a 0.22 μm filter was used before the supernatant was injected into the GC–MS system. Analysis was performed using aTRACE1300-TSQ9000 gas chromatography (Thermo Fisher Scientific, Waltham, MA, USA). The extracts were separated using an Agilent HP-FFAP (25 m × 0.32 μm × 0.50 mm) for analysis of SCFAs (acetic acid, propionic acid, butyric acid, and isovaleric acid). Two microliters of the sample were manually injected into a split/splitless inlet (in split mode, 30:1), which was kept at 250 °C. The oven program was set at an initial temperature of 100 °C, and increased to 160 °C at a rate of 5 °C/min for 2 min. The temperatures of the transfer line and electron impact ion source were set at 2800 °C and 250 °C, respectively. The solvent delay was set at 3.5 min. The chromatogram was mapped in SIM mode with the characteristic ions of SCFAs obtained.

### 2.10. Analysis of Gut Microbiota

The DNA was extracted with the TGuide S96 Magnetic Soil DNA Kit (Model: DP812; Tiangen Biotech Co., Ltd., Beijing, China) according to the manufacturer’s instructions. The DNA concentration of the samples was measured with the Qubit dsDNA HS Assay Kit and Qubit 4.0 Fluorometer (Invitrogen, Thermo Fisher Scientific, St. Bend, OR, USA). After amplification, PCR products were detected by electrophoresis with 1.8% agarose (Beijing Bomei Fuxin Technology Co., Ltd., Beijing, China). The 27F: AGRGTTTGATYNTGGCTCAG and 1492R: TASGGHTACCTTGTTASGACTT universal primer set was used to amplify the full-length 16S rRNA gene from the genomic DNA extracted from each sample [27]. SMRTbell libraries were prepared from the amplified DNA by SMRTbell Express Template Prep Kit 2.0 according to the manufacturer’s instructions (Pacific Biosciences, Menlo Park, CA, USA). Purified SMRTbell libraries from the pooled and barcoded samples were sequenced on a single PacBio Sequel II 8M cell using the Sequel II Sequencing kit 2.0. UCHIME software was used to filter chimeric sequences in the obtained clean tags [28]. The resulting sequences with an identity more significant than 97% were classified as operational taxonomic units (OTUs) [29]. Bioinformatic analysis was performed on the cloud platform (https://www.bioincloud.tech, accessed on 22 November 2021) of Shenzhen Weikemeng Technology Group Co., Ltd. (Shenzhen, China).

### 2.11. Statistical Analysis

Statistical analyses were performed using IBM SPSS Statistics 26 (International Business Machines Corporation, Armonk, NY, USA), and the statistical results were expressed as mean ± standard deviation (S.D.). Analysis of variance (ANOVA) with Duncan’s range was used to compare the differences between groups, and *p* < 0.05 was considered a significant difference. The sequencing data are analyzed on the cloud platform of Weikemeng Technology Group Co., Ltd. (https://www.bioincloud.tech/, accessed on 22 November 2021).

## 3. Results

### 3.1. Body Weight and Fasting Blood Glucose

Weight loss is a typical symptom of T2DM [30,31,32]. The body weight of mice in the M, P, V, R, CL, FL, FM, and FH groups was significantly decreased (*p* < 0.05) compared with the N and FN groups before administration. After the four-week intervention, the weight loss of the V, CL, FL, and FM groups was significantly suppressed (*p* < 0.05) compared with the M group (Figure 2A). The FBG was significantly decreased, by 56.08%, 52.82%, and 45.18% in the FL, FM, and FH groups (*p* < 0.05), respectively, and the effect of the FBG decrease was better in the FL, FM, and FH groups than in the P, V, R, and CL groups (Figure 2B). These results indicated that the FL, FM, and FH groups were shown to be more effective than the P, V, R, and CL groups in suppressing weight loss and decreasing FBG levels in T2DM mice.

### 3.2. Oral Glucose Tolerance

The glucose levels rose rapidly in all groups within the first 30 min, peaked at 30 min, and then began to decline slowly. After 120 min, glucose in the N and FN groups returned to normal levels; the glucose levels of the P, V, R, CL, FL, FM, and FH groups were significantly decreased (*p* < 0.05) compared with the M group; and the FL group had a better effect than the other groups (Figure 3A). The AUC of the P, V, R, CL, FL, FM, and FH groups was significantly decreased (*p* < 0.05) compared with the M group, of which the FL group had a most significant decrease, of 26.80% (Figure 3B).

### 3.3. Effects of FRRT on Liver and Kidney

FRRT can repair and improve the liver and kidney function of T2DM mice. After four weeks of FRRT intervention, the liver and kidney coefficients of the FL and FM groups were significantly decreased (*p* < 0.05), and there were no significant differences in liver and kidney coefficients between the P, V, R, and CL groups (*p* > 0.05) compared with the M group (Figure 4A,B). The abnormality of liver and kidney biochemical indexes is closely related to T2DM [33,34]. The TG levels of the P, V, R, CL, FL, FM, and FH groups were significantly decreased, by 23.48%, 36.00%, 49.28%, 55.66%, 38.09%, 63.55%, and 31.62% (Figure 4C, *p* < 0.05), respectively; the TC levels of the V, R, CL, FL, and FM groups were significantly decreased, by 20.30%, 39.36%, 41.52%, 27.85%, and 41.15% (Figure 4D, *p* < 0.05), respectively; the HDL-C levels of the P, V, R, CL, FL, FM, and FH groups were significantly increased (Figure 4E, *p* < 0.05); and the LDL-C levels of the P, V, CL, and FM groups were significantly decreased (Figure 4F, *p* < 0.05) compared with the M group. The BUN and creatinine levels are used as indicators to judge renal function. The BUN levels of the P, V, R, CL, FL, FM, and FH groups were significantly decreased, by 5.75%, 4.33%, 16.09%, 16.46%, 14.70%, 10.98%, and 9.04% (*p* < 0.05), respectively, and the creatinine levels of the P, CL, FL, FM, and FH groups were significantly decreased, by 21.02%, 18.45%, 23.28%, 35.30%, and 21.78% (*p* < 0.05), respectively, compared with the M group (Figure 4G,H). These results indicated that the FM group (medium-dose FRRT) showed better liver and kidney biochemical indexes than other groups. The FM group was significantly better than the P group in TG, TC, BUN, and creatinine levels (*p* < 0.05); the FM group was slightly lower than the P group in HDL-C levels (*p* < 0.05); and there was no significant difference between the FM and the P groups in LDL-C levels (*p* > 0.05).

### 3.4. Histological Traits of Liver and Kidney

The structure of hepatocytes in the N group was well-defined, regular, well-arranged, and evenly distributed. However, in the M group the hepatocytes were vacuolated and necrotic, and many fat particles infiltrated the liver. After four weeks of intervention, the infiltration of fat granules and the degree of vacuolation and necrosis of hepatocytes in the P, V, and FM groups were alleviated (Figure 5A). Mesangium enlargement, tubulointerstitial fibrosis, glomerulosclerosis, and vacuolar degeneration of glomerular epithelial cells were observed in the M group. After four weeks of intervention, the glomerular interstitial fibrosis, epithelial vacuoles, and glomerular atrophy were improved in the R, FL, and FM groups (Figure 5B).

### 3.5. Effects of FRRT on Oxidative Stress

Oxidative stress is a pathogenesis of T2DM [35,36]. Compared with the M group, the GSH-PX levels of the P, V, CL, FL, FM, and FH groups were significantly increased, by 49.63%, 44.69%, 13.15%, 46.52%, 18.52%, and 17.04% (Figure 6A, *p* < 0.05), respectively. The CAT levels of the P, V, FL, FM, and FH groups were significantly increased, by 165.22%, 41.30%, 417.39%, 123.91%, and 260.87% (Figure 6B, *p* < 0.05), respectively. The SOD levels in each group were significantly increased (Figure 6C, *p* < 0.05). The T-AOC levels in the FL and FH groups were significantly increased, by 49.63% and 44.69% (Figure 6D, *p* < 0.05), respectively. The MDA levels in the FL and FM groups were significantly decreased, by 67.22% and 68.46% (Figure 6E, *p* < 0.05), respectively. FRRT could improve the oxidative stress induced by T2DM to some extent and enhanced the antioxidant capacity of T2DM mice.

### 3.6. Short-Chain Fatty Acids

The gut microbiota benefit humans by producing SCFAs through carbohydrate fermentation, and insufficient SCFAs production is related to T2DM [37]. SCFAs can effectively improve metabolic diseases such as obesity, T2DM, and metabolic syndrome [38]. The levels of acetic acid, propionic acid, butyric acid, and isovaleric acid in the FN group were significantly increased (*p* < 0.05) compared with group N. The levels of acetic acid in the P, V, R, CL, FL, and FM groups were significantly increased, by 24.90%, 80.92%, 11.11%, 26.69%, 9.81%, and 142.28% (*p* < 0.05), respectively; the levels of propionic acid in the V, R, CL, FM, and FH groups were significantly increased, by 643.97%, 69.98%, 127.44%, 428.59%, and 81.39% (*p* < 0.05), respectively, compared with the M group (Figure 7A). The levels of butyric acid in the P, V, R, CL, FL, FM, and FH groups were significantly increased, by 508.37%, 3622.56%, 572.38%, 1040.03%, 104.86%, 1968.66%, and 836.35% (*p* < 0.05), respectively; the levels of isovaleric acid in the P, R, CL, FL, FM, and FH groups were significantly increased, by 1116.19%, 71.08%, 603.80%, 121.74%, 81.04%, and 101.93% (*p* < 0.05), respectively, compared with the M group (Figure 7B). The FM group (medium-dose FRRT) was significantly better than the P, R, CL, FL, and FH groups in the acetic acid, propionic acid, and butyric acid levels (*p* < 0.05).

### 3.7. Composition Analysis of the Gut Microbiota

FRRT affected the diversity and abundance of the gut microbiota in T2DM mice. The gut microbiota of each group of mice were sequenced, tags with more than 97% similarity were clustered, and each OTU obtained was considered as the same species [29]. OTUs varied among intervention groups, with the FN group having the highest OTUs, followed by the FM group (Figure 8A). The alpha diversity of species within the gut microbiome is a common outcome of interest that is examined in microbiome research. It contains two sub-constructs: richness and evenness of the species composition in the sample. The Chao1 index is commonly used to indicate microbial species richness, and the Shannon and Simpson indexes represent microbial species diversity [39]. The Chao1 and Shannon indexes were significantly higher in the FN group than in the N group (Figure 8B,C, *p* < 0.05), indicating that the abundance and diversity of the gut microbiota were significantly increased (*p* < 0.05) in healthy mice after FRRT intervention. The Simpson indexes in the P, V, R, CL, FL, and FM groups were significantly decreased (*p* < 0.05) compared to the M group, indicating that the diversity of the gut microbiota was significantly increased (Figure 8D, *p* < 0.05) in the P, V, R, CL, FL, and FM groups. Beta diversity compares the composition of microbial communities between different samples, indicating the differences between microbial communities, which is measured by comparing the distance values between samples; the closer the distance between the samples is, the more similar the species composition is. The sample distance matrix is analyzed using dimension-reducing sorting through principal component analysis (PCA) and non-metric multidimensional scaling (NMDS). Then, the sample relationship is intuitively judged on the three-dimensional plane [40,41]. These results indicated that the P, R, FL, and FH groups are relatively close, indicating that metformin hydrochloride, *Lactobacillus paracasei* SR10-1 bacterial, and FRRT have similar effects on the gut microbiota of T2DM mice. After the intervention of FRRT in the FN group, the gut microbiota were significantly different from the other groups (Figure 8E,F, *p* < 0.05).

The phylum-level gut microbiota were mainly composed of Firmicutes, Proteobacteria, Bacteroidetes, Verrucomicrobia, Cyanobacteria, Deferribacterota, Actinobacteria, Tenericutes, and TM7. The relative abundance of Firmicutes in the P (84.46%), V (94.65%), CL (89.37%), and FM (83.33%) groups was significantly increased (*p* < 0.05) compared with the M group (74.93%). The relative abundance of Proteobacteria in the M group (13.40%) was significantly increased (*p* < 0.05) compared with the N group (3.38%). After FRRT intervention, the relative abundance of Proteobacteria returned to 3.26% in the FL group and 5.02% in the FM group (Figure 9A), indicating that the V, CL, FL, and FM groups reduced the ratio of Firmicutes to Proteobacteria (Figure 9B).

For the family-level gut microbiota, the relative abundance of Lachnospiraceae in the R, FL, FM, and FH groups was significantly increased (*p* < 0.05) compared with the M group. Enterobacteriaceae is thought to be associated with overweight and obesity [42]. The relative abundance of Enterobacteriaceae in the M group (11.18%) was significantly increased (*p* < 0.05) compared with the N group (0.02%). After the intervention, the relative abundance of Enterobacteriaceae in the V (1.12%), CL (0.43%), FL (1.18%), and FM (0.00%) groups was significantly decreased (*p* < 0.05). Veillonellaceae is a common Gram-negative anaerobe that can cause endogenous intestinal infection as a conditional pathogen. The relative abundance of Veillonellaceae in the gut microbiota of diabetic children was significantly higher than that of healthy children [43,44]. In the present study, Veillonellaceae only appeared in the M group, and the relative abundance of 8.28% indicated that high-fat and high-sugar feed-induced diabetic mice increased the number of pathogenic bacteria. However, Veillonellaceae did not appear in the FL, FM, and FH groups after the FRRT intervention (Figure 10A).

For the genus-level gut microbiota, *Akkermansia* is a novel probiotic that degrades mucin in the human gut and is inversely associated with obesity, diabetes, inflammation, and metabolic disorders. Decreased abundance of *Akkermansia* leads to the impairment of insulin secretion and glucose homeostasis in T2DM [45]. The relative abundance of *Akkermansia* in the R (13.59%), FL (17.95%), FM (11.40%), and FH (16.01%) groups was significantly increased (*p* < 0.05) compared with the M group (8.80%). *Klebsiella*, a conditioned pathogen that parasitizes the respiratory or intestinal tract of animals, is one of the pathogens causing pneumonia in humans. Various strains in the genus *Klebsiella* have evolved to become a major clinical and public health threat worldwide [46]. The relative abundance of *Klebsiella* in the M group (10.20%) was significantly increased (*p* < 0.05) compared with the N group (0.00%). After the FRRT intervention, the relative abundance of *Klebsiella* reduced to 1.18% in the FL group and 0.00% in the FM group. *Allobaculum* was positively correlated with the expression of angiopoietin-like protein 4 [47], and the relative abundance of *Allobaculum* was 0.02% in the N group, whereas that of the FN group was significantly increased after FRRT intervention, with a relative abundance of 5.62% (*p* < 0.05). *Phascolarctobacterium* only appeared in the M group (8.28%), and *Phascolarctobacterium* was an essential biomarker in obese patients with T2DM and might be a potential therapeutic target [48] (Figure 10B).

### 3.8. Correlation between Oxidative Stress and Gut Microbiota

Linear discriminant analysis (LDA) Effect Size (LEfSe) is an algorithm to robustly identify features that are statistically different among biological classes [49]. Zhang et al. found that the relative abundance of Verrucomicrobiaceae in T2DM mice was significantly decreased [50]. In the present study, the relative abundance of Verrucomicrobiaceae and *Akkermansia* in the FL group was significantly increased (*p* < 0.05) after the FRRT intervention (Figure 11A,B). Weight was significantly positively correlated with *Odoribacter* (*p* < 0.001), *Coprobacillus* (*p* < 0.001), *Helicobacter* (*p* < 0.01), *Bifidobacterium* (*p* < 0.01), *Parabacteroides* (*p* < 0.05), *Sutterella* (*p* < 0.05), *Bacteroides* (*p* < 0.05), and *Psychrobacter* (*p* < 0.05), and significantly negatively correlated with *Escherichia* (*p* < 0.001), *Klebsiella* (*p* < 0.001), *Weissella* (*p* < 0.05), *Phascolarctobacterium* (*p* < 0.05), and *Akkermansia* (*p* < 0.05). GSH-PX was significantly positively correlated with *Dorea* and *Lactobacillus* (*p* < 0.05), and significantly negatively correlated with *Coprococcus* (*p* < 0.001). SOD was significantly positively correlated with *Ruminococcus* (*p* < 0.01), and significantly negatively correlated with *Parabacteroides* (*p* < 0.01), *Coprococcus* (*p* < 0.01), and *Phascolarctobacterium* (*p* < 0.01). CAT was significantly positively correlated with *Ruminococcus* (*p* < 0.01) and *Dorea* (*p* < 0.01), and significantly negatively correlated with *Phascolarctobacterium* (*p* < 0.05). T-AOC was significantly positively correlated with *Ruminococcus* (*p* < 0.05), and significantly negatively correlated with *Turicibacter* (*p* < 0.05) and *Phascolarctobacterium* (*p* < 0.05) (Figure 11C).

## 4. Discussion

RRT is rich in bioactive substances, such as vitamin C, polysaccharides, polyphenols, and SOD, often used in the research and development of functional foods [13]. Polysaccharides [17] and polyphenols [22] in RRT have been proven to have hypoglycemic effects. The previous research of our team found that RRT fruit vinegar had an ameliorating effect on dyslipidemia and other phenomena in obese mice [25]. However, the relationship between FRRT and T2DM was unclear. In the present study, FRRT was used to intervene in STZ-induced T2DM mice. Body weight, blood glucose, the liver coefficient, the kidney coefficient, oxidative stress, SCAFs, and gut microbiota were measured in both T2DM and healthy mice.

Weight loss is an early symptom of T2DM, mainly due to the relative insulin deficiency caused by pancreatic β-cell dysfunction and insulin resistance in target organs [1]. The body does not fully utilize glucose for energy production, leading to increased fat and protein breakdown, over-consumption, negative nitrogen balance, and even weight loss [31]. In the present study, the weight loss of mice in the FL and FM groups was significantly suppressed (Figure 2A, *p* < 0.05). FBG in the FL and FM groups was significantly decreased (Figure 2B, *p* < 0.05). Glucose tolerance refers to the body’s ability to regulate blood sugar levels. In oral glucose tolerance tests, the FL and FM groups improved the glucose tolerance of T2DM mice (Figure 3A,B). The FL (low-dose FRRT) and FM (medium-dose FRRT) groups showed significantly better results than the P group (metformin hydrochloride) in inhibiting weight loss and controlling FBG and glucose tolerance.

Patients with diabetes are often accompanied by dyslipidemia, leading to increased TG, TC, LDL-C, BUN, and creatinine levels and decreased HDL-C levels [48,51]. Dyslipidemia in T2DM mice was ameliorated after metformin hydrochloride (P group), RRTJ (CL group), low-dose FRRT (FL group), and medium-dose FRRT (FM group) interventions, and the effect of medium-dose FRRT was significantly superior to that of metformin hydrochloride, RRTJ, and low-dose FRRT. Fermented juice had significantly better effects on the regulation of blood lipids, oxidative stress, and antidiabetes than unfermented juice [32,52].

Oxidative stress is a pathogenesis of T2DM. The levels of GSH-PX, CAT, SOD, and T-AOC were significantly decreased (*p* < 0.05), and the levels of MDA were significantly increased (*p* < 0.05) in T2DM mice (Figure 6). The SOD and ascorbic acid in RRT can balance oxidative stress in humans. Oxidative stress is caused by a physiological imbalance between levels of antioxidants and oxidants (free radicals and reactive species) [20]. The GSH-PX and SOD levels of T2DM mice in the CL group were significantly increased (Figure 6A,C, *p* < 0.05), and the SOD levels of T2DM mice in the R group were significantly increased (Figure 6C, *p* < 0.05). After RRTJ was fermented by *Lactobacillus paracasei* SR10-1 (FRRT), the levels of GSH-PX, CAT, SOD, and T-AOC had significantly increased (*p* < 0.05), indicating the synergistic effect of *Lactobacillus paracasei* SR10-1, and RRTJ significantly improved oxidative stress in T2DM mice (Figure 6).

Observational findings achieved during the past two decades suggest that gut microbiota are essential for human health [4]. Humans benefit from the production of SCFAs by the gut microbiota through carbohydrate fermentation, and it stands to reason that insufficient production of SCFAs is associated with T2DM [37]. Su et al. reported that dietary fiber from RRT residue could increase the production of SCFAs by colon fermentation in vitro [14]. The SCFAs of T2DM mice in the FM group were more significantly increased (*p* < 0.05) than in the P, R, and CL groups (Figure 7), and it also suggests that *Lactobacillus paracasei* SR10-1 fermentation enhanced the functionality of RRTJ, and the organic combination of this strain and RRTJ enhanced the efficiency of FRRT. The healthy and T2DM mice had significantly increased (*p* < 0.05) SCFAs production after medium-dose FRRT intervention (Figure 7), an intriguing result that suggests that, for a healthy population, FRRT can benefit gut microbiota as a functional food. The SCFAs levels of vitamin C intervention were significantly increased, especially propionic acid and butyric acid [53]. Propionic acid can enhance glucose-stimulated insulin release and maintain β-cell mass by inhibiting apoptosis [40]. Butyric acid can increase insulin sensitivity, inhibit liver gluconeogenesis, and improve host sugar metabolism and insulin resistance [41]. And in the present study, the propionic acid and butyric acid levels in T2DM mice of vitamin C intervention (vitamin C: 1000 mg/100 g) were more significantly increased (*p* < 0.05) than of medium-dose FRRT intervention (vitamin C: 375.65 ± 7.39 mg/100 g).

Gut microbiota are closely related to human health [54,55,56]. It has been confirmed that T2DM is closely associated with the gut microbiota. Das et al. first reported significant differences in the diversity and abundance of gut microbiota at the phylum and genus levels between diabetic retinopathy patients and healthy individuals [57]. High-fat and high-sugar diet-induced T2DM leads to strong alterations in the composition of the gut microbiota [58]. In the present study, the relative abundance of Enterobacteriaceae that was considered to be associated with overweight and obesity [42] in the M group (11.18%) was significantly increased (*p* < 0.05) compared with the N group (0.02%). Veillonellaceae is a common Gram-negative anaerobe that can cause endogenous intestinal infections as a conditional pathogen, which occurs only in the M group (8.28%) (Figure 10A). *Phascolarctobacterium* was significantly negatively correlated with weight (*p* < 0.05), SOD (*p* < 0.01), CAT (*p* < 0.05), and T-AOC (Figure 11C, *p* < 0.05), which was consistent with the study by Wang et al., who found that *Phascolarctobacterium* was an essential biomarker in obese patients with T2DM [48]. *Klebsiella* is present in the gut microbiota of T2DM mice, suggesting that people with diabetes are more susceptible to *Klebsiella peneumoniae* [59]. The butyric acid content in the cecum was significantly increased due to the stimulation of Lachnospiraceae [60]. *Akkermansia* is a novel probiotic that degrades mucin in the human gut and is inversely associated with obesity, diabetes, inflammation, and metabolic disorders. *Akkermansia* is reduced, impairing insulin secretion and glucose homeostasis in T2DM [45]. In the correlation analysis, *Akkermansia* was significantly negatively correlated (*p* < 0.05) with weight (Figure 11C). RRT treatment can reverse the gut microbiota disturbance from a high-fat diet [61]. Metformin hydrochloride intervention increased the relative abundance of Firmicutes; vitamin C intervention increased the relative abundance of Firmicutes and decreased the relative abundance of Enterobacteriaceae; *Lactobacillus paracasei* SR10-1 intervention increased the relative abundance of Lachnospiraceae and *Akkermansia*; RRTJ intervention increased the relative abundance of Firmicutes and decreased the relative abundance of Enterobacteriaceae; low-dose FRRT intervention increased the relative abundance of Lachnospiraceae and *Akkermansia* and decreased the relative abundance of Proteobacteria, Enterobacteriaceae, Veillonellaceae, *Klebsiella*, and *Phascolarctobacterium*; medium-dose FRRT intervention increased the relative abundance of Firmicutes, Lachnospiraceae, *Akkermansia*, and *Allobaculum* and decreased the relative abundance of Proteobacteria, Enterobacteriaceae, Veillonellaceae, *Klebsiella*, and *Phascolarctobacterium*; and high-dose FRRT intervention increased the relative abundance of Lachnospiraceae and *Akkermansia*.

The medium-dose FRRT intervention (FM group) showed better antidiabetic effects than metformin hydrochloride (P group), vitamin C (V group), *Lactobacillus paracasei* SR10-1 (R group), RRTJ (CL group), low-dose FRRT (FL group), and high-dose FRRT (FH group) interventions. Therefore, the imbalance in the gut microbiota of T2DM mice was effectively alleviated after FRRT intervention, and medium-dose FRRT intervention was more effective than other interventions in gut microbiota regulation. This finding also provides a reference for FRRT to explore the relationship between the gut microbiome and T2DM and other metabolic syndromes. However, it is not clear whether FRRT improves T2DM by one or more components.

## 5. Conclusions

In summary, this study reveals the regulatory effects of metformin hydrochloride, vitamin C, *Lactobacillus paracasei* SR10-1, RRTJ, low-dose FRRT, medium-dose FRRT, and high-dose FRRT on oxidative stress, dyslipidemia, and gut microbiota in high-sugar and high-fat diets and STZ-induced T2DM. The synergistic effect of *Lactobacillus paracasei* SR10-1 and RRTJ fermented FRRT with a better antidiabetic effect. The medium-dose FRRT (FM group) reverses dyslipidemia, relieves oxidative stress, increases SCFAs levels, and regulates the composition of gut microbiota better than the other intervention groups. The results show that the improvement of body weight and oxidative stress in T2DM mice after supplementation by medium-dose FRRT are highly correlated with the *Odoribacter*, *Coprobacillus*, *Coprococcus*, *Ruminococcus*, *Dorea*, *Lactobacillus*, *Helicobacter*, *Bifidobacterium*, *Parabacteroides*, *Phascolarctobacterium*, *Sutterella*, *Bacteroides*, *Psychrobacter*, *Escherichia*, *Klebsiella*, *Weissella*, and *Akkermansia*. This study demonstrated that FRRT has excellent potential for dietary prevention of T2DM and related chronic diseases. The medium-dose of FRRT supplementation has beneficial health effects on healthy and T2DM mice.

## Figures and Tables

**Figure 1 foods-12-03233-f001:**
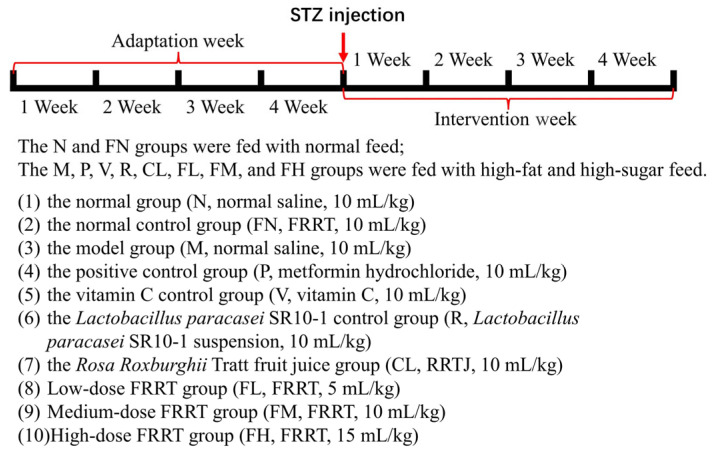
Overview of the T2DM model procedure.

**Figure 2 foods-12-03233-f002:**
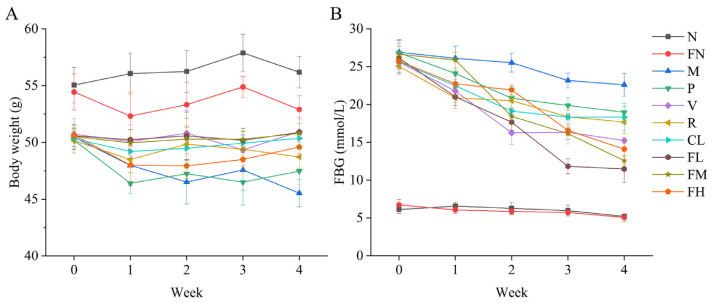
Effect of FRRT on the body weight and FBG of STZ-induced T2DM mice. (**A**) body weight; (**B**) FBG. All data are presented as the mean ± standard deviation. N: the normal group (normal saline, 10 mL/kg), FN: the normal control group (FRRT, 10 mL/kg), M: the model group (normal saline, 10 mL/kg), P: the positive control group (metformin hydrochloride, 10 mL/kg), V: the vitamin C control group (vitamin C, 10 mL/kg), R: the *Lactobacillus paracasei* SR10-1 control group (*Lactobacillus paracasei* SR10-1 suspension, 10 mL/kg), CL: the *Rosa roxburghii* Tratt fruit juice group (RRTJ, 10 mL/kg), FL: the low-dose FRRT group (FRRT, 5 mL/kg), FM: the medium-dose FRRT group (FRRT, 10 mL/kg), and FH: the high-dose FRRT group (FRRT, 15 mL/kg).

**Figure 3 foods-12-03233-f003:**
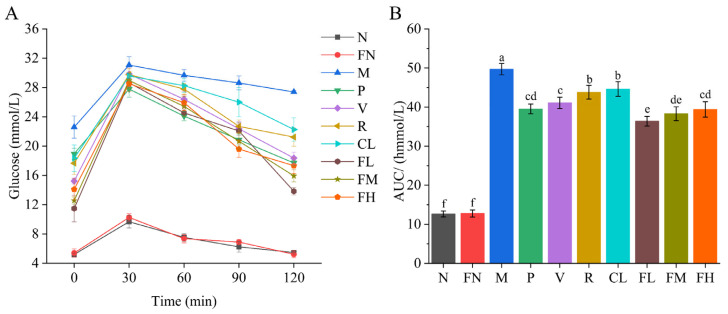
Effect of FRRT on oral glucose tolerance and the AUC of STZ-induced T2DM mice. (**A**) oral glucose tolerance; (**B**) AUC. Different superscript characters indicate differences significantly among groups (*p* < 0.05). All data are presented as the mean ± standard deviation. N: the normal group (normal saline, 10 mL/kg), FN: the normal control group (FRRT, 10 mL/kg), M: the model group (normal saline, 10 mL/kg), P: the positive control group (metformin hydrochloride, 10 mL/kg), V: the vitamin C control group (vitamin C, 10 mL/kg), R: the *Lactobacillus paracasei* SR10-1 control group (*Lactobacillus paracasei* SR10-1 suspension, 10 mL/kg), CL: the *Rosa roxburghii* Tratt fruit juice group (RRTJ, 10 mL/kg), FL: the low-dose FRRT group (FRRT, 5 mL/kg), FM: the medium-dose FRRT group (FRRT, 10 mL/kg), and FH: the high-dose FRRT group (FRRT, 15 mL/kg).

**Figure 4 foods-12-03233-f004:**
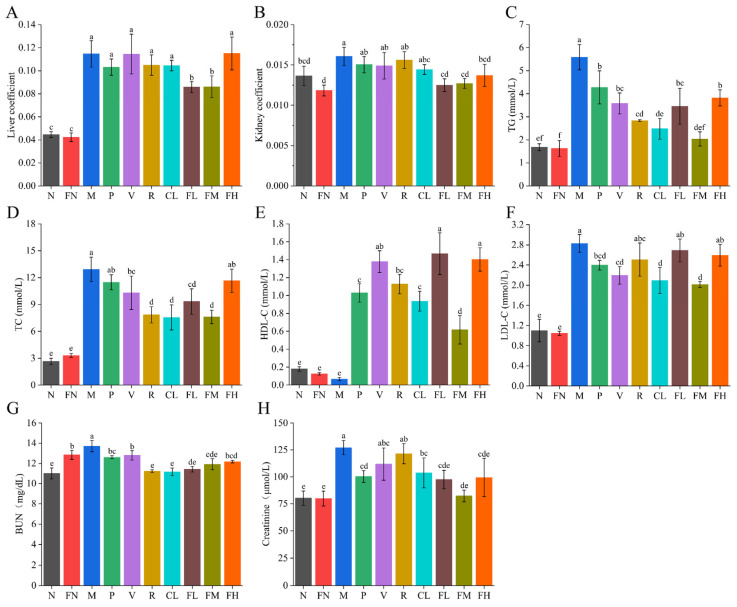
Effects of FRRT on the liver and kidneys of STZ-induced T2DM mice. (**A**) liver coefficient; (**B**) kidney coefficient; effect of FRRT on liver biochemical indexes of T2DM mice: (**C**) TG, (**D**) TC, (**E**) HDL-C, and (**F**) LDL-C; effect of FRRT on kidney biochemical indexes of T2DM mice: (**G**) BUN and (**H**) Creatinine. Different superscript characters indicate differences significantly among groups (*p* < 0.05). All data are presented as the mean ± standard deviation. N: the normal group (normal saline, 10 mL/kg), FN: the normal control group (FRRT, 10 mL/kg), M: the model group (normal saline, 10 mL/kg), P: the positive control group (metformin hydrochloride, 10 mL/kg), V: the vitamin C control group (vitamin C, 10 mL/kg), R: the *Lactobacillus paracasei* SR10-1 control group (*Lactobacillus paracasei* SR10-1 suspension, 10 mL/kg), CL: the *Rosa roxburghii* Tratt fruit juice group (RRTJ, 10 mL/kg), FL: the low-dose FRRT group (FRRT, 5 mL/kg), FM: the medium-dose FRRT group (FRRT, 10 mL/kg), and FH: the high-dose FRRT group (FRRT, 15 mL/kg).

**Figure 5 foods-12-03233-f005:**
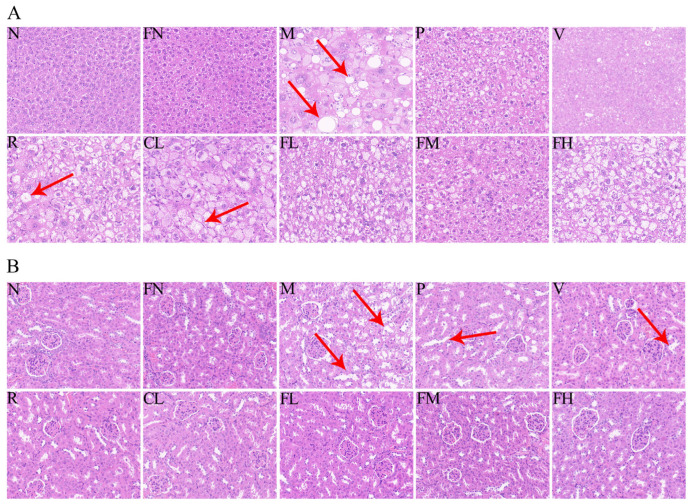
Effects of FRRT on the liver and kidneys of STZ-induced T2DM mice. (**A**) H&E staining of the liver (×40 magnification); (**B**) H&E staining of the kidneys (×40 magnification). Different superscript characters indicate differences significantly among groups (*p* < 0.05). All data are presented as the mean ± standard deviation. N: the normal group (normal saline, 10 mL/kg), FN: the normal control group (FRRT, 10 mL/kg), M: the model group (normal saline, 10 mL/kg), P: the positive control group (metformin hydrochloride, 10 mL/kg), V: the vitamin C control group (vitamin C, 10 mL/kg), R: the *Lactobacillus paracasei* SR10-1 control group (*Lactobacillus paracasei* SR10-1 suspension, 10 mL/kg), CL: the *Rosa roxburghii* Tratt fruit juice group (RRTJ, 10 mL/kg), FL: the low-dose FRRT group (FRRT, 5 mL/kg), FM: the medium-dose FRRT group (FRRT, 10 mL/kg), and FH: the high-dose FRRT group (FRRT, 15 mL/kg).

**Figure 6 foods-12-03233-f006:**
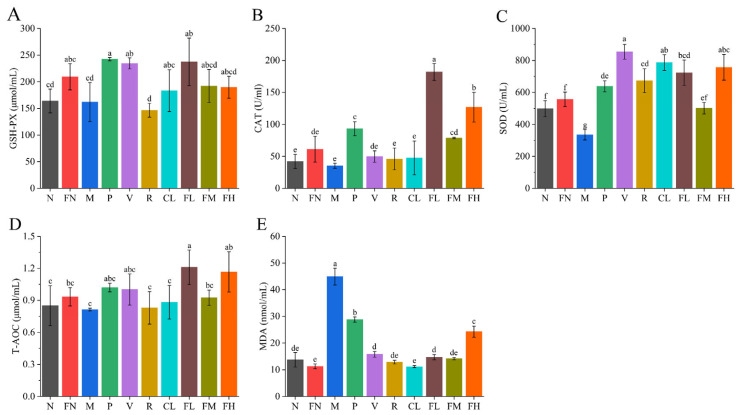
Effects of FRRT on oxidative stress of STZ-induced T2DM mice. (**A**) GSH-PX; (**B**) CAT; (**C**) SOD; (**D**) T-AOC; and (**E**) MDA. Different superscript characters indicate differences significantly among groups (*p* < 0.05). All data are presented as the mean ± standard deviation. N: the normal group (normal saline, 10 mL/kg), FN: the normal control group (FRRT, 10 mL/kg), M: the model group (normal saline, 10 mL/kg), P: the positive control group (metformin hydrochloride, 10 mL/kg), V: the vitamin C control group (vitamin C, 10 mL/kg), R: the *Lactobacillus paracasei* SR10-1 control group (*Lactobacillus paracasei* SR10-1 suspension, 10 mL/kg), CL: the *Rosa roxburghii* Tratt fruit juice group (RRTJ, 10 mL/kg), FL: the low-dose FRRT group (FRRT, 5 mL/kg), FM: the medium-dose FRRT group (FRRT, 10 mL/kg), and FH: the high-dose FRRT group (FRRT, 15 mL/kg).

**Figure 7 foods-12-03233-f007:**
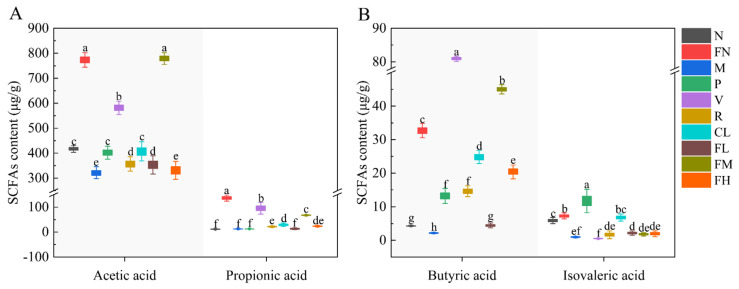
Effects of FRRT on SCFAs of STZ-induced T2DM mice. (**A**) acetic acid and propionic acid; (**B**) butyric acid and isovaleric acid. Different superscript characters indicate differences significantly among groups (*p* < 0.05). All data are presented as the mean ± standard deviation. N: the normal group (normal saline, 10 mL/kg), FN: the normal control group (FRRT, 10 mL/kg), M: the model group (normal saline, 10 mL/kg), P: the positive control group (metformin hydrochloride, 10 mL/kg), V: the vitamin C control group (vitamin C, 10 mL/kg), R: the *Lactobacillus paracasei* SR10-1 control group (*Lactobacillus paracasei* SR10-1 suspension, 10 mL/kg), CL: the *Rosa roxburghii* Tratt fruit juice group (RRTJ, 10 mL/kg), FL: the low-dose FRRT group (FRRT, 5 mL/kg), FM: the medium-dose FRRT group (FRRT, 10 mL/kg), and FH: the high-dose FRRT group (FRRT, 15 mL/kg).

**Figure 8 foods-12-03233-f008:**
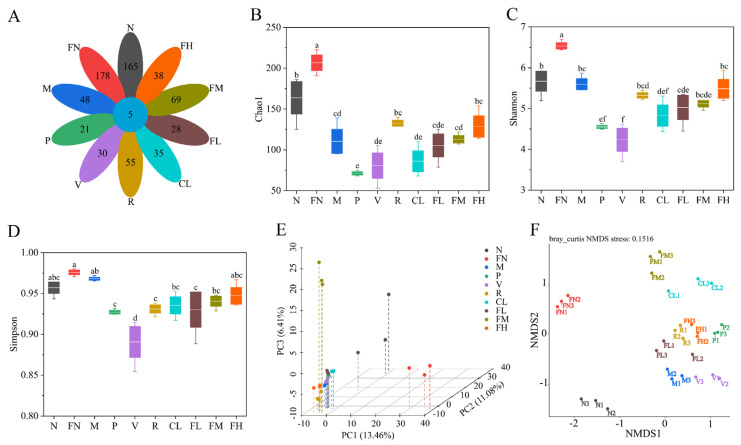
Effects of FRRT on the gut microbiota of STZ-induced T2DM mice. (**A**) operational taxonomic units; alpha diversity of gut microbiota in T2DM mice: (**B**) Chao1 index, (**C**) Shannon index, and (**D**) Simpson index; beta diversity of gut microbiota in T2DM mice: (**E**) principal component analysis and (**F**) non-metric multidimensional scaling. Different superscript characters indicate differences significantly among groups (*p* < 0.05). All data are presented as the mean ± standard deviation. N: the normal group (normal saline, 10 mL/kg), FN: the normal control group (FRRT, 10 mL/kg), M: the model group (normal saline, 10 mL/kg), P: the positive control group (metformin hydrochloride, 10 mL/kg), V: the vitamin C control group (vitamin C, 10 mL/kg), R: the *Lactobacillus paracasei* SR10-1 control group (*Lactobacillus paracasei* SR10-1 suspension, 10 mL/kg), CL: the *Rosa roxburghii* Tratt fruit juice group (RRTJ, 10 mL/kg), FL: the low-dose FRRT group (FRRT, 5 mL/kg), FM: the medium-dose FRRT group (FRRT, 10 mL/kg), and FH: the high-dose FRRT group (FRRT, 15 mL/kg).

**Figure 9 foods-12-03233-f009:**
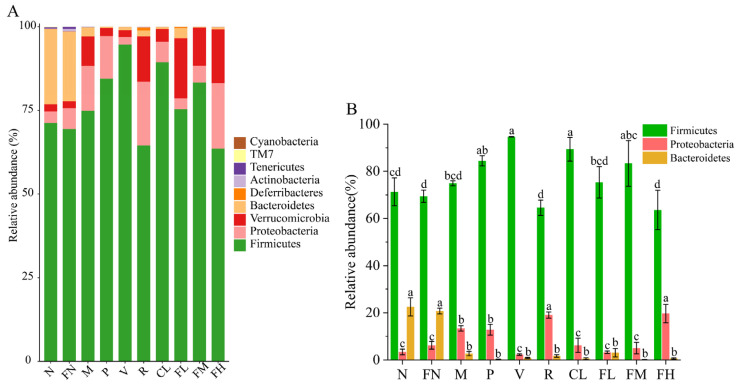
Species composition of gut microbiota in T2DM mice: (**A**) phylum-level abundance of gut microbiota, (**B**) relative abundance of Firmicutes, *Bacteroidetes*, and Proteobacteria. Each phylum is represented by a unique color. N: the normal group (normal saline, 10 mL/kg), FN: the normal control group (FRRT, 10 mL/kg), M: the model group (normal saline, 10 mL/kg), P: the positive control group (metformin hydrochloride, 10 mL/kg), V: the vitamin C control group (vitamin C, 10 mL/kg), R: the *Lactobacillus paracasei* SR10-1 control group (*Lactobacillus paracasei* SR10-1 suspension, 10 mL/kg), CL: the *Rosa roxburghii* Tratt fruit juice group (RRTJ, 10 mL/kg), FL: the low-dose FRRT group (FRRT, 5 mL/kg), FM: the medium-dose FRRT group (FRRT, 10 mL/kg), and FH: the high-dose FRRT group (FRRT, 15 mL/kg).

**Figure 10 foods-12-03233-f010:**
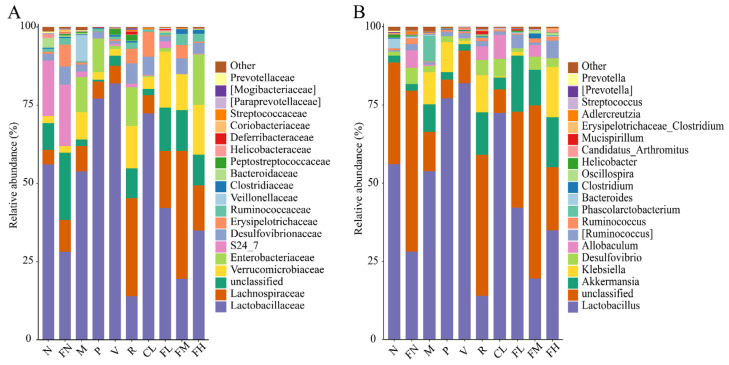
Species composition of gut microbiota in T2DM mice: (**A**) family-level abundance of gut microbiota, and (**B**) genus-level abundance of gut microbiota. Each family or genus is represented by a unique color. N: the normal group (normal saline, 10 mL/kg), FN: the normal control group (FRRT, 10 mL/kg), M: the model group (normal saline, 10 mL/kg), P: the positive control group (metformin hydrochloride, 10 mL/kg), V: the vitamin C control group (vitamin C, 10 mL/kg), R: the *Lactobacillus paracasei* SR10-1 control group (*Lactobacillus paracasei* SR10-1 suspension, 10 mL/kg), CL: the *Rosa roxburghii* Tratt fruit juice group (RRTJ, 10 mL/kg), FL: the low-dose FRRT group (FRRT, 5 mL/kg), FM: the medium-dose FRRT group (FRRT, 10 mL/kg), and FH: the high-dose FRRT group (FRRT, 15 mL/kg).

**Figure 11 foods-12-03233-f011:**
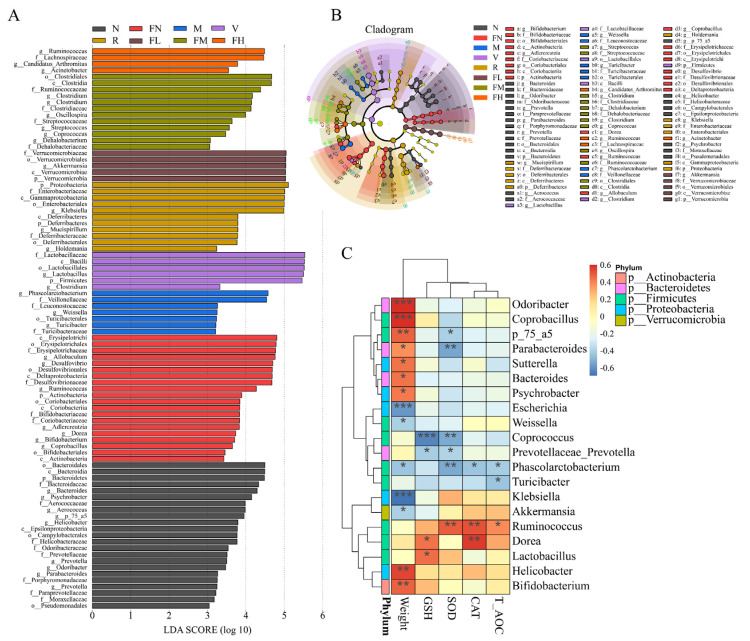
(**A**) linear discriminant analysis, (**B**) linear discriminant analysis Effect Size, and (**C**) a heat map of the relationship between genus-level gut microbiota species and phenotypes. Note: * *p* < 0.05, ** *p* < 0.01, and *** *p* < 0.001. N: the normal group (normal saline, 10 mL/kg), FN: the normal control group (FRRT, 10 mL/kg), M: the model group (normal saline, 10 mL/kg), P: the positive control group (metformin hydrochloride, 10 mL/kg), V: the vitamin C control group (vitamin C, 10 mL/kg), R: the *Lactobacillus paracasei* SR10-1 control group (*Lactobacillus paracasei* SR10-1 suspension, 10 mL/kg), CL: the *Rosa roxburghii* Tratt fruit juice group (RRTJ, 10 mL/kg), FL: the low-dose FRRT group (FRRT, 5 mL/kg), FM: the medium-dose FRRT group (FRRT, 10 mL/kg), and FH: the high-dose FRRT group (FRRT, 15 mL/kg).

## Data Availability

The datasets generated for this study are available on request to the corresponding author.

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
