# Peer review of "Effect and Correlation of Rosa roxburghii Tratt Juice Fermented by Lactobacillus paracasei SR10-1 on Oxidative Stress and Gut Microflora Dysbiosis in Streptozotocin (STZ)-Induced Type 2 Diabetes Mellitus Mice"

_foods, 2023, doi:10.3390/foods12173233_

Round 1

Reviewer 1 Report

Comments and Suggestions for Authors

The manuscript is about effect of fermented Rosa roxburghii Tratt juice on oxidative stress and gut microbiota in streptozotocin-induced Type 2 diabetes mellitus mice. It is well organized and informative to relevant readers. However, the authors should consider the following things to improve the quality of the manuscript.

Line 72: 120 -> One hundred twenty

Line 92: FBG ≥ 11.1mmol/L -> FBG ≥ 11.1 mmol/L

M&M Sample Collection: “At the day before the end of the experiment, the mice were placed in a sterile mouse cage, and fresh feces samples of mice were collected in sterile Eppendorf tubes.” -> Did not the authors sample the feces from mice just before intervention? It might help to understand the basal microbiota compositions of the mice.

Line 103: 12h -> 12 h

Line 147: 2 μL -> Two microliter

Line 154: (Tiangen Biotech (Beijing) Co., Ltd, China, and model: DP812) -> (Model: DP812; Tiangen Biotech Co., Ltd., Beijing, China)

Line 159: Ltd -> Ltd.

Line 176: data is -> data are

Fig. 1: Why is so different in body weight and FBG between normal and high glucose and fat diet mice at 0-week point?

Line 201: 120min -> 120 min

Fig. 2: Why did show more effectiveness in FL than FM and FH?

Line 260: “After four weeks intervention, the glomerular interstitial fibrosis, epithelial vacuoles, and glomerular atrophy in the R, FL, and FM groups (Figure 4B)” -> Please complete the sentence.

Fig. 3-6. Based on the Figure 3-5, it doesn’t show dose-dependent manner. Can the authors explain the phenomena?

Line 358: “indicating that V, CL, FL, and FM groups reduced the ratio of Firmicutes to Proteobacteria” -> V, CL, FL, and FM groups increased(?)

Line 376: “Akkermansia is reduced, impairing insulin secretion and glucose homeostasis in T2DM [45].” -> Please rephrase the sentence.

Line 393: “(B) Shannon index, and (C) Simpson index” -> (C) Shannon index, and (D) Simpson index

Line 401: “FL: Low-dose FRRT control group (FRRT, 5 mL/kg), FM: Medium-dose FRRT control group (FRRT, 10 mL/kg), FH: High-dose FRRT control group (FRRT, 15 mL/kg).” -> FL: Low-dose FRRT group (FRRT, 5 mL/kg), FM: Medium-dose FRRT group (FRRT, 10 mL/kg), FH: High-dose FRRT group (FRRT, 15 mL/kg).

Figure 7: Based on the data, Shannon and Simpson indices of M group are comparably higher than other groups including positive control. Can the authors justify that the samples improve gut microbiota in STZ-induced T2DM mice?

Comments on the Quality of English Language

It is well organized and clear.

Author Response

Thank you very much for your suggestions. The modifications are highlighted in Word.

Reviewer 2 Report

Comments and Suggestions for Authors

The research article “Effect and Correlation of Rosa roxburghii Tratt Juice Fermented by Lactobacillus paracasei SR10-1 on oxidative stress and Gut Microflora Disorder in Streptozotocin (STZ)-induced Type 2 Diabetes Mellitus Mice” deals with an interesting topic and is a well-written and structured article. It is easy to follow the methods and results and the findings of the study represents valuable information. However, written English should be revised by an English language specialist. I only have some minor requests for revision:

·         Please follow the rule of abbreviations in the text. For the first time, use both a full and abbreviated form, next time, just use an acronym (i. e. TG, TC, LDL-C, BUN, etc.)

·         Line 4; “gut microflora dysbiosis” instead of “gut microflora disorder”

·         Authors should format the references according to the journal instructions.

Comments on the Quality of English Language

written English should be revised by an English language specialist

Author Response

(The authors gave the same response as above.)

Reviewer 3 Report

Comments and Suggestions for Authors

This manuscript is a rigorous study of the components of a fermented extract of Rosa Roxburghii Tratt in type 2 diabetes in a mouse model of HFD and STZ. Different doses of the fermented extract were used and were compared with the effects of metformin and vitamin C. The overall design is sound, the data is interesting and very significant to human health. However, several issues need to be address before the manuscript is considered for publication, as listed below:

1.     The description of the experiment is confusing. The groups should be explained better, adding HFD to the corresponding group could help, like FL-HFD, the model control should be just HFD, the positive control could be Met, and so on. A diagram of the different treatment and timing should be added for clarity.

2.     For Fig 1, a significance symbol should be added for every data point that is significantly different from the controls.

3.     Was the diet purified? What is the formulation of the control low fat diet?

4.     For the reduction in body weight, was this caused by the HFD or a combination of HFD and STZ? It would be important to show body weight changes during the initial 4-week period of HFD before STZ injection.

5.     Add a description of the letters “a, b, c, d, e” in Fig 2B to the figure legend. The same for the rest of the figures.

6.     Indicate by arrows the histopathology observed in Fig 4. A quantification of the scores should be provided.

7.     Was the fermented juice analyzed for other molecules such as polyphenols? For most of the measured outcomes the fermented fruit performed better than vitamin C, suggesting that the effect of this fermented extract is not mediated by its vitamin C content. Other mechanisms should be discussed. It would be interesting if authors can provide the polyphenol composition of the extract before and after fermentation.

8.     Add a limitation section.

Comments on the Quality of English Language

It would be beneficial if the manuscript is revised by an English speaking scientist 

Author Response

(The authors gave the same response as above.)
